# SIAMESE ATTENTION NETWORKS

## ABSTRACT

Attention operators have been widely applied on data of various orders and dimensions such as texts, images, and videos. One challenge of applying attention operators is the excessive usage of computational resources. This is due to the usage of dot product and softmax operator when computing similarity scores. In this work, we propose the Siamese similarity function that uses a feed-forward network to compute similarity scores. This results in the Siamese attention operator (SAO). In particular, SAO leads to a dramatic reduction in the requirement of computational resources. Experimental results show that our SAO can save 94% memory usage and speed up the computation by a factor of 58 compared to the regular attention operator. The computational advantage of SAO is even larger on higher-order and higher-dimensional data. Results on image classification and restoration tasks demonstrate that networks with SAOs are as effective as models with regular attention operator, while significantly outperform those without attention operators.

## 1 INTRODUCTION

Deep learning networks with attention operators have demonstrated great capabilities of solving challenging problems in various tasks such as computer vision (Xu et al., 2015; Lu et al., 2016), natural language processing (Bahdanau et al., 2015; Vaswani et al., 2017), and network embedding (Veličković et al., 2017). Attention operators are capable of capturing long-range relationships and brings significant performance boosts (Li et al., 2018; Malinowski et al., 2018). The application scenarios of attention operators range from 1-D data like texts to high-order and high-dimensional data such as images and videos. However, attention operators suffer from the excessive usage of computational resources when applied on high-order or high-dimensional data. The memory and computational cost increases dramatically with the increase of input orders and dimensions. This prevents attention operators from being applied in broader scenarios. To address this limitation, some studies focus on reducing spatial sizes of inputs such as down-sampling input data (Wang et al., 2018) or attending selected part of data (Huang et al., 2018). However, such kind of methods inevitably results in information and performance loss.

In this work, we propose a novel and efficient attention operator known as Siamese attention operator (SAO) to dramatically reduce the usage of computational resources. We observe that the excessive computational resource usage is mainly caused by the similarity function and coefficients normalization function used in attention operators. To address this limitation, we propose the Siamese similarity function that employs a feed-forward network to compute similarity scores. By applying the same network to both input vectors, Siamese similarity function processes the symmetry property. By using Siamese similarity function to compute similarity scores, we propose the Siamese attention operator, which results in a significant saving on computational resources. Based on the Siamese attention operator, we design a family of efficient modules, which leads to our compact deep models known as Siamese attention networks (SANets). Our SANets significantly outperform other state-of-the-art compact models on image classification tasks. Experiments on image restoration tasks demonstrate that our methods are efficient and effective in general application scenarios.

## 2 BACKGROUND AND RELATED WORK

In this section, we describe the attention operator that has been widely applied in various tasks and on various types of data including texts, images, and videos.

## 2.1 ATTENTION OPERATOR

The inputs of an attention operator include three matrices; those are a query matrix $\boldsymbol{Q} = [\mathbf{q}_1, \mathbf{q}_2, \cdots, \mathbf{q}_m] \in \mathbb{R}^{d \times m}$ with each $\mathbf{q}_i \in \mathbb{R}^d$, a key matrix $\boldsymbol{K} = [\mathbf{k}_1, \mathbf{k}_2, \cdots, \mathbf{k}_n] \in \mathbb{R}^{d \times n}$ with each $\mathbf{k}_i \in \mathbb{R}^d$, and a value matrix $\boldsymbol{V} = [\mathbf{v}_1, \mathbf{v}_2, \cdots, \mathbf{v}_n] \in \mathbb{R}^{p \times n}$ with each $\mathbf{v}_i \in \mathbb{R}^p$. To compute the response of each query vector $\boldsymbol{q}_i$, the attention operator calculates similarity scores between $\boldsymbol{q}_i$ and each key vector $\boldsymbol{k}_j$ using a similarity function. Frequently used similarity functions include dot product (Luong et al., 2015), concatenation (Bahdanau et al., 2015), Gaussian function, and embedded Gaussian function. It has been shown that dot product is the most effective one (Wang et al., 2018), which computes $\text{sim}(\boldsymbol{q}_i, \boldsymbol{k}_j) = \boldsymbol{k}_j^T \boldsymbol{q}_i$. After the normalization with a softmax operator, the response is computed by taking a weighted sum over value vectors $\sum_{j=1}^N \boldsymbol{v}_j \text{softmax}(\boldsymbol{k}_j^T \boldsymbol{q}_i)$. For all query vectors, the attention operator computes

$$\boldsymbol{O} = \boldsymbol{V} \text{softmax}(\boldsymbol{K}^T \boldsymbol{Q}), \tag{1}$$

where $\text{softmax}(\cdot)$ is the column-wise softmax operator. The matrix multiplication between $\boldsymbol{K}^T$ and $\boldsymbol{Q}$ computes a intermediate output matrix $\boldsymbol{E}$ that stores similarity scores between each query vector $\boldsymbol{q}_i$ and each key vector $\boldsymbol{k}_j$. The column-wise softmax operator normalizes $\boldsymbol{E}$ and makes every column sum to 1. Multiplication between $\boldsymbol{V}$ and the normalized $\boldsymbol{E}$ gives the output $\boldsymbol{O} \in \mathbb{R}^{p \times m}$. Self-attention operator (Vaswani et al., 2017; Devlin et al., 2018) is a special case of the attention operator with $\boldsymbol{Q} = \boldsymbol{K} = \boldsymbol{V}$. In practice, we usually firstly perform linear transformations on input matrices. For notation simplicity, we use original input matrices in following discussions. The computational cost of the operations in Eq. (1) is $O(m \times n \times (d + p))$, and the memory required to store the intermediate output $\boldsymbol{E}$ is $O(m \times n)$. If $m = n$ and $d = p$, the time and space complexities of Eq. (1) are $O(n^2 \times d)$ and $O(n^2)$, respectively.

The matrix multiplication order in Eq. (1) is determined by the softmax operator, which acts as the normalization function. Wang et al. (2018) proposed to use scaling by $1/N$ as the normalization function on similarity scores. By this, the response of $\boldsymbol{q}_i$ is calculated $\frac{1}{N} \sum_{j=1}^N \boldsymbol{v}_j \boldsymbol{k}_j^T \boldsymbol{q}_i$. The attention operator using scaling by $1/N$ computes all responses as:

$$\boldsymbol{O} = \frac{1}{N}(\boldsymbol{V} \boldsymbol{K}^T)\boldsymbol{Q}. \tag{2}$$

By computing $\boldsymbol{V} \boldsymbol{K}^T$ first, the time and space complexities of Eq. (2) are $O(Nd^2)$ and $O(d^2)$, respectively. When $N > d$, this saves computational resources compared to the attention operator in Eq. (1). In practice, we usually have $N > d$ in some parts of a neural network, especially on high-order data.

## 2.2 ATTENTION OPERATORS ON HIGH-ORDER DATA

Non-local operators (Wang et al., 2018) are essentially self-attention operators on high-order data like images and videos. Take 2-D data as an example, the input to a non-local operator is an image, which can be represented as a third-order tensor $\mathcal{X} \in \mathbb{R}^{h \times w \times c}$. Here, $h$, $w$, and $c$ denote the height, width, and number of channels, respectively. The non-local operator converts the tensor into a matrix $\boldsymbol{X}_{(3)} \in \mathbb{R}^{c \times hw}$ by unfolding along mode-3 (Kolda & Bader, 2009). Then the matrix is fed into an attention operator by setting $\boldsymbol{Q} = \boldsymbol{K} = \boldsymbol{V} = \boldsymbol{X}_{(3)}$. The output of the attention operator is converted back to a third-order tensor that is used as the final output. A challenging problem of non-local operators is the excessive usage of computational resources. If $h = w$, the time and space complexity of the non-local operator is $O(h^4 \times d)$ and $O(h^4)$, respectively. The computational cost becomes even bigger on higher-order data like videos. The excessive usage of computational resources limits the application of attention operators in broader scenarios.

## 3 SIAMESE ATTENTION NETWORKS

In this work, we propose a learnable similarity function known as the Siamese similarity function. This function uses a single-layer feed-forward network to compute similarity scores. Based on Siamese similarity function, we propose the Siamese attention operator, which dramatically reduces computational cost. We also describe how to build Siamese attention networks using this operator.

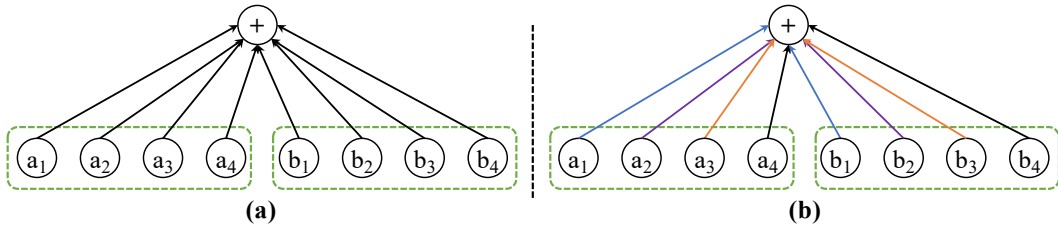

Figure 1: Illustration of the similarity function using a single-layer feed-forward network (a) and the Siamese similarity function (b). In (a), different input elements use different weights. In (b), two input vectors share the same network. The lines with the same color indicate weights sharing.

### 3.1 SIAMESE SIMILARITY FUNCTION

We analyze the problem of learning a similarity function given two vectors. To learn a similarity function, we employ a single-layer feed-forward neural network. Given two vectors $\boldsymbol{a} \in \mathbb{R}^d$ and $\boldsymbol{b} \in \mathbb{R}^d$, the similarity score is computed using a trainable vector $\boldsymbol{w} \in \mathbb{R}^{2d}$ as:

$$\text{sim}_{\boldsymbol{w}}(\boldsymbol{a}, \boldsymbol{b}) = [\boldsymbol{a}^T, \boldsymbol{b}^T]\boldsymbol{w} = \sum_{i=1}^{d} a_i \times w_i + b_i \times w_{d+i} = \boldsymbol{a}^T \boldsymbol{w}_a + \boldsymbol{b}^T \boldsymbol{w}_b, \quad (3)$$

where $\boldsymbol{w} = [\boldsymbol{w}_a^T, \boldsymbol{w}_b^T]^T$ with $\boldsymbol{w}_a \in \mathbb{R}^d$ and $\boldsymbol{w}_b \in \mathbb{R}^d$. Here, we ignore the bias term for notation simplicity. We consider the similarity function defined in Eq. (3) as two feed-forward networks that process two vectors separately. The similarity score is the sum of outputs from two different networks.

Unlike distance metrics, the non-negativity or triangle inequality do not need to hold from similarity functions. But we usually expect similarity measures to be symmetric, which means it outputs the same similarity score when two input arguments are swapped. Apparently, the similarity function defined by Eq. (3) does not have this property. To retain the symmetry property, we employ the same network while using both vectors to compute the similarity score. This leads to our proposed Siamese similarity function (Sia-sim), which follows the principle of Siamese networks (Bromley et al., 1994; Bertinetto et al., 2016). The Siamese similarity function computes the similarity score between $\boldsymbol{a}$ and $\boldsymbol{b}$ as:

$$\text{Sia-sim}_{\boldsymbol{w}}(\boldsymbol{a}, \boldsymbol{b}) = \sum_{i=1}^{d} (a_i + b_i) \times w_i = (\boldsymbol{a} + \boldsymbol{b})^T \boldsymbol{w}, = \text{Sia-sim}_{\boldsymbol{w}}(\boldsymbol{b}, \boldsymbol{a}) \quad (4)$$

where $\boldsymbol{w} \in \mathbb{R}^d$ is a trainable parameter vector. Although the time complexity of computing Sia-sim is the same as that of dot product, we show that Sia-sim leads to a very efficient attention operator in Section 3.2. Figure 1 provides an illustration of the similarity functions defined in Eq. (3) and Eq. (4).

### 3.2 SIAMESE ATTENTION OPERATOR

We describe the Siamese attention operator in the context of 1-D data, but it can be easily applied on high-order data by unfolding them into matrices. In this case, the inputs to an attention operator are $\boldsymbol{Q} \in \mathbb{R}^{d \times N}$, $\boldsymbol{K} \in \mathbb{R}^{d \times N}$, and $\boldsymbol{V} \in \mathbb{R}^{d \times N}$. We replace the similarity function in the attention operator by our Siamese similarity function, leading to the Siamese attention operator (SAO). Given a query vector $\boldsymbol{q}_i$ in $\boldsymbol{Q}$, SAO computes the response $\boldsymbol{o}_i$ as:

$$\boldsymbol{o}_i = \frac{1}{N} \sum_{j=1}^{N} \boldsymbol{v}_j (\boldsymbol{q}_i + \boldsymbol{k}_j)^T \boldsymbol{w} \qquad = \frac{1}{N} \sum_{j=1}^{N} (\boldsymbol{v}_j \boldsymbol{q}_i^T \boldsymbol{w} + \boldsymbol{v}_j \boldsymbol{k}_j^T \boldsymbol{w})$$

$$= \left( \frac{1}{N} \sum_{j=1}^{N} \boldsymbol{v}_j \right) \boldsymbol{q}_i^T \boldsymbol{w} + \frac{1}{N} \left( \sum_{j=1}^{N} \boldsymbol{v}_j \boldsymbol{k}_j^T \right) \boldsymbol{w} \quad = \overline{\boldsymbol{v}} \boldsymbol{w}^T \boldsymbol{q}_i + \frac{1}{N} \boldsymbol{V} \boldsymbol{K}^T \boldsymbol{w}, \quad (5)$$

where $\overline{\boldsymbol{v}} = \frac{1}{N} \sum_{j=1}^{N} \boldsymbol{V}_{:j} \in \mathbb{R}^d$. SAO computes responses of all query vectors as:

$$\boldsymbol{O} = \overline{\boldsymbol{v}} \boldsymbol{w}^T \boldsymbol{Q} + \frac{1}{N} \boldsymbol{V} \boldsymbol{K}^T \boldsymbol{w} \mathbf{1}_N^T, \quad (6)$$

Table 1: Comparisons among three attention operators in terms of time and space complexities. Attn and $\text{Attn}_{1/N}$ denote the regular attention operators using softmax and scaling by $1/N$ for similarity scores normalization, respectively.

| Operator | Computation | Time Complexity | Space Complexity |
|---|---|---|---|
| Attn | $\boldsymbol{V}\text{softmax}(\boldsymbol{K}^T\boldsymbol{Q})$ | $O(N^2 \times d)$ | $O(N^2)$ |
| $\text{Attn}_{1/N}$ | $\frac{1}{N}(\boldsymbol{V}\boldsymbol{K}^T)\boldsymbol{Q}$ | $O(N \times d^2)$ | $O(d^2)$ |
| SAO | $\overline{\boldsymbol{v}}\boldsymbol{w}^T\boldsymbol{Q} + \frac{1}{N}\boldsymbol{V}\boldsymbol{K}^T\boldsymbol{w}\boldsymbol{1}_N^T$ | $O(N \times d)$ | $O(N \times d)$ |

where $\boldsymbol{1}_N$ is a vector of ones of size $N$.

Note that we use $\boldsymbol{1}_N$ here to make it mathematically precise. In practice, the term $\frac{1}{N}\boldsymbol{V}\boldsymbol{K}^T\boldsymbol{w}$ is the same to all query vectors. This means we only need to compute it once and share it for the computation of all responses. By computing $\boldsymbol{K}^T\boldsymbol{w}$ first, the time complexity of this term is $O(N \times d)$. Similarly, the time complexity for computing the first term in Eq. (6) is $O(N \times d)$. Thus, the overall time complexity of SAO is $O(N \times d)$. Notably, when $\boldsymbol{Q} = \boldsymbol{K}$, we can save the computational cost by computing either $\boldsymbol{w}^T\boldsymbol{Q}$ or $\boldsymbol{K}^T\boldsymbol{w}$. Table 1 provides the comparison of SAO and other attention operators. It can be seen from the comparison results that our SAO can significantly save computational resources compared to other attention operators.

In Eq. (5), the first response term $\overline{\boldsymbol{v}}\boldsymbol{w}^T\boldsymbol{q}_i$ changes as the query vector $q_i$, which we call a local response term. The second term $\frac{1}{N}\boldsymbol{V}\boldsymbol{K}^T\boldsymbol{w}$ is the same for all query vectors, which is a global response term. The local response term provides customized information to query vectors, while the global response term may include global information for SAO. In the experimental study part, we demonstrate the importance of the global response term to SAO.

## 3.3 SIAMESE ATTENTION MODULES

Attention models have been shown to be effective in various fields. However, they are not widely applied in compact neural networks to date primarily due to their excessive computational cost. Based on our efficient SAO, it is possible to integrate attention operators in compact convolutional neural networks (CNNs) such as MobileNetV2 (Sandler et al., 2018). In this section, we design a family of efficient Siamese attention modules (SAMs) that integrate our SAO with convolutional operators. These modules can be used for designing compact CNNs.

**BaseModule:** Bottleneck blocks with inverted residuals are main components in MobileNetV2. Each bottleneck consists of three layers; those are an $1\times1$ convolutional layer for expansion, a $3\times3$ depth-wise convolutional layer, and another $1\times1$ convolutional layer. The hyper-parameters of this module include the expansion factor $r$ and the stride $s$. Given the input tensor $\boldsymbol{\mathcal{X}}^{(\ell)} \in \mathbb{R}^{h\times w\times c}$ for the $\ell$th block, the first $1\times1$ convolutional layer expands it into $(r-1)c$ feature maps. We then concatenate the output with the input, resulting in $\tilde{\boldsymbol{\mathcal{X}}}^{(\ell)} \in \mathbb{R}^{h\times w\times rc}$. The depth-wise convolutional layer with stride $s$ outputs $rc$ feature maps $\bar{\boldsymbol{\mathcal{X}}}^{(\ell)} \in \mathbb{R}^{\frac{h}{s}\times\frac{w}{s}\times rc}$. Finally, the last $1\times1$ convolutional layer outputs $d$ feature maps $\boldsymbol{\mathcal{Y}}^{(\ell)} \in \mathbb{R}^{\frac{h}{s}\times\frac{w}{s}\times d}$. A skip connection is applied between $\boldsymbol{\mathcal{X}}^{(\ell)}$ and $\boldsymbol{\mathcal{Y}}^{(\ell)}$ when $c = d$ and $s = 1$. The BaseModule is illustrated in Figure 2 (a).

**AttnModule:** We propose to integrate our SAO into the BaseModule, resulting in AttnModule as shown in Figure 2 (b). Before the last $1\times1$ convolutional layer, we add a new parallel path by SAO, which outputs $c$ feature maps. An average pooling layer with stride $s$ is followed when $s > 1$, resulting in $\bar{\boldsymbol{\mathcal{X}}}_a^{(\ell)} \in \mathbb{R}^{\frac{h}{s}\times\frac{w}{s}\times c}$. In the original path, both the first $1\times1$ convolutional layer and the depth-wise convolutional layer output $(r-1)c$ feature maps $\bar{\boldsymbol{\mathcal{X}}}_b^{(\ell)} \in \mathbb{R}^{\frac{h}{s}\times\frac{w}{s}\times(r-1)c}$. We concatenate and feed them into the last $1\times1$ convolutional layer, which outputs $d$ feature maps $\boldsymbol{\mathcal{Y}}^{(\ell)} \in \mathbb{R}^{\frac{h}{s}\times\frac{w}{s}\times d}$. In SAOs, we only apply linear transformation on the value matrix $\boldsymbol{V}$ to limit the computational cost and the number of trainable parameters. The original path extracts locality-based features, while the new path using SAO computes global features. In this way, AttnModule is capable of capturing both local and global information. We add a skip connection between inputs and outputs of SAO to enable better feature reuse and gradient back-propagation (He et al., 2016a).

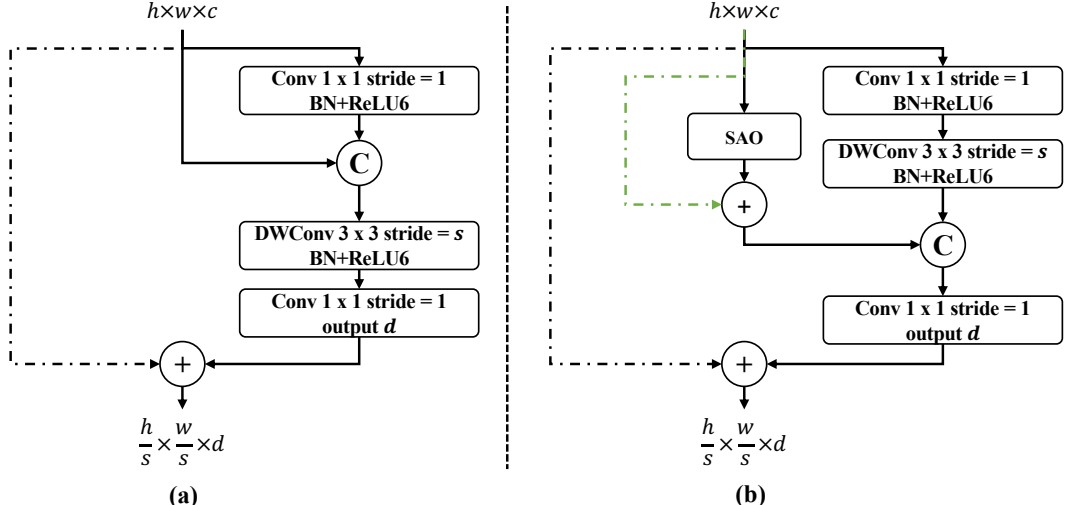

Figure 2: Architectures of the BaseModule (a) and AttnModule (b) as introduced in Section 3.3. The skip connection between inputs and outputs is used when $s = 1$ and $c = d$. The one between inputs and outputs of SAO is used when $s = 1$.

### 3.4 SIAMESE ATTENTION NETWORKS

In this section, we describe the model architecture of a compact convolutional neural network using our proposed SAOs and modules in Section 3.3, leading to Siamese attention networks (SANets). We follow the basic architecture of MobileNetV2 (Sandler et al., 2018) and replace inverted residual blocks with our SAMs. The details of network architecture are described in Table 9 in the appendix. Following the practices in Wang et al. (2018), we use AttnModules to replace the blocks with input spatial sizes of $28 \times 28$, $14 \times 14$, and $7 \times 7$. For the rest of the blocks, we use BaseModule with the same expansion factors. The SANet built on SAMs saves about 0.03 million parameters and 12 million MAdd compared to MobileNetV2. We show that SANets with SAOs significantly outperform MobileNetV2 while using less computational resources.

## 4 EXPERIMENTAL STUDIES

In this section, we evaluate our methods using image classification and restoration tasks. We firstly evaluate our SAO in terms of computational resource usage by comparing with regular attention operators. We compare our designed SANets with compact CNNs on image classification tasks using the ImageNet ILSVRC 2012 dataset (Russakovsky et al., 2015). We conduct ablation studies to investigate the benefits of our SAO and the trainable parameters in it. We also evaluate our methods in general application scenarios using image restoration tasks.

### 4.1 COMPARISON OF COMPUTATIONAL EFFICIENCY

The theoretical analysis in Section 3.2 shows that our SAO can achieve significant efficiency advantages over regular attention operator especially on high-order and high-dimensional data. In this section, we use simulated experiments to evaluate the efficiency advantage of SAO over other attention operators. We build networks using different attention operators. The network is composed of a single attention operator to eliminate the influence of other factors. We apply the TensorFlow profile tool (Abadi et al., 2016) and report the required memory and CPU time on 2-D simulated data. To fully evaluate the efficiency of attention operators, we vary the number of channels and spatial sizes of the input data.

The comparison results is summarized in Table 2. On the simulated data with the size of $56^2 \times 256$, our SAO achieves 94.65% memory saving and 58.21 times speed-up compared to the regular attention operator. Our propose SAO can significantly save computational resources. The efficiency advantage

Table 2: Comparisons between the regular attention operator, the regular attention operator with a pooling operation (Wang et al., 2018), and our proposed SAO in terms of the number of parameters, number of MAdd, memory usage, and CPU inference time on simulated data of different sizes. The input sizes are given in the format of "spatial sizes$^2 \times$ number of input channels". "Attn" denotes the regular attention operator. "Attn+Pool" denotes the regular attention operator which employs $2 \times 2$ pooling operations on $K$ and $V$ input matrices to reduce required computational resources.

| Input | Operator | MAdd | Memory | Saving | Time | Speedup |
|---|---|---|---|---|---|---|
| $14^2 \times 64$ | Attn | 4.92m | 20.97MB | 0.00% | 20.47ms | 1.00× |
| | Attn+Pool | 1.23m | 8.39MB | 59.99% | 5.07ms | 4.04× |
| | Attn$_{1/N}$ | 1.61m | 5.24MB | 75.01% | 6.97ms | 2.94× |
| | SAO | **0.05m** | **4.29MB** | **79.54%** | **3.09ms** | **6.62×** |
| $28^2 \times 128$ | Attn | 157.35m | 301.99MB | 0.00% | 450.22ms | 1.00× |
| | Attn+Pool | 39.34m | 100.66MB | 66.67% | 119.55ms | 3.77× |
| | Attn$_{1/N}$ | 25.69m | 37.75MB | 87.50% | 70.52ms | 6.38× |
| | SAO | **0.40m** | **33.88MB** | **88.78%** | **31.76ms** | **14.18×** |
| $56^2 \times 256$ | Attn | 5,035.26m | 5.04GB | 0.00% | 12.92s | 1.00× |
| | Attn+Pool | 1,258.82m | 1.34GB | 73.38% | 4.82s | 2.68× |
| | Attn$_{1/N}$ | 411.04m | 0.28GB | 94.34% | 1.02s | 12.66× |
| | SAO | **3.21m** | **0.26GB** | **94.65%** | **0.22s** | **58.21×** |

of SAO over regular attention operators increases as the increase of spatial and dimension sizes. When comparing with other attention operators, our SAO is shown to be the most computationally efficient attention operator. The simulated results show that our SAO is an efficient attention operator that can operate high-dimensional and high-order data by consuming very few computational resources.

## 4.2 RESULTS ON IMAGE CLASSIFICATION

Based on our efficient SAO, we build a family of compact CNNs in Section 3.4 for image classification tasks. To evaluate the effectiveness of our SAO and SANets, we compare our models with other compact CNNs (Howard et al., 2017; Sandler et al., 2018; Zhang et al., 2017; Gao et al., 2018) on the ImageNet ILSVRC 2012 image classification dataset. It has been serving as the benchmark dataset, especially for image classification tasks. The dataset contains 1.2 million images for training, 50 thousand images for validation, and 50 thousand images for testing. We provide the experimental setups on image classification tasks in the appendix.

Table 3: Comparisons between SANet and other CNNs in terms of the top-1 accuracy, the number of trainable parameters, and MAdd on the ImageNet validation set.

| Model | Top-1 | Params | MAdd |
|---|---|---|---|
| GoogleNet | 0.698 | 6.8m | 1550m |
| VGG16 | 0.715 | 128m | 15300m |
| SqueezeNet | 0.575 | 1.3m | 833m |
| MobileNetV1 | 0.706 | 4.2m | 569m |
| ChannelNet-v1 | 0.705 | 3.7m | 407m |
| ShuffleNet 1.5x | 0.715 | 3.4m | 292m |
| MobileNetV2 | 0.720 | 3.47m | 300m |
| **SANet** (ours) | **0.730** | 3.44m | **288m** |

We compare our SANets with other compact CNNs and report the top-1 accuracy, the number of parameters, and MAdd in Table 3. MobileNetV2 (Sandler et al., 2018) is the previous state-of-the-art model in terms of the computational cost and model performance. Compared to MobileNetV2, our SANet significantly outperforms MobileNetV2 by a margin of 1% with a smaller number of parameters. By using our SAO, SANet can achieve new state-of-the-art performance with limited computational resources. Considering that we only make minor modifications from the architecture of MobileNetV2, the performance boost is significant. Compared to the module using regular convolutional layers, our proposed module uses SAO to obtain global features and avoids the excessive usage of computational resources. Our SAO successfully overcomes the limitations of regular attention operator and applies attention mechanism on high-order and high-dimensional data with significant performance boost. Next, we will show our SAO is as effective as regular attention operator but can dramatically reduce the usage of computational resources.

### 4.3 COMPARISON WITH REGULAR ATTENTION OPERATORS

We have shown that our SAO has significant efficiency advantages over regular attention operators in Section 4.1. In this section, we conduct experiments to compare our SAO with Squeeze-and-Excite block (Hu et al., 2018) and regular attention operators based on SANets using image classification tasks. Besides the regular attention operator, we also consider the one defined in Eq. (2) using scaling by $1/N$ for coefficients normalization, and the one with a pooling operation as in Wang et al. (2018). In the attention operator with pooling operations, $2 \times 2$ pooling operations are applied to key matrix $\boldsymbol{K}$ and value matrix $\boldsymbol{V}$, leading to reduced spatial sizes. To ensure fair comparisons, we replace all SAOs in SANets with attention operators, attention operators with pooling operations, and attention operators defined in Eq. (2), denoted as AttnNet, AttnNet+Pool, and AttnNet$_{1/N}$, respectively. We also compare the actual computational usage among MobileNetV2, SANet and AttnNet.

Table 4: Comparisons between SANet, the network using the same architecture as SANet with Squeeze-and-Excite block (denoted as SENet) or with regular attention operators (denoted as AttnNet), the AttnNet using regular attention operators with a pooling operation (AttnNet+Pool), and the AttnNet with regular attention operators using scaling by 1/N (denoted as AttnNet$_{1/N}$) in terms of the top-1 accuracy, the number of total parameters, and MAdd on the ImageNet validation set.

| Model | Top-1 | Params | MAdd |
|---|---|---|---|
| SENet | 0.724 | 3.44m | 294m |
| AttnNet | 0.730 | 3.44m | 365m |
| AttnNet+Pool | 0.729 | 3.44m | 300m |
| AttnNet$_{1/N}$ | 0.730 | 3.44m | 312m |
| SANet | 0.730 | 3.44m | 288m |

We summarize the comparison results in Table 4 and Table 5. We can observe from the results that our SANet using SAOs achieves the same performance as AttnNet with regular attention operators but using significantly less computational resources. Compared to MobileNetV2, our SANet uses slightly extra resources but achieves significantly better performances. The results demonstrate that our SAO is as effective as the regular attention operator while dramatically reduces computational costs. The better performance of SANet over AttnNet+pool indicates that our SAO is more efficient and effective compared to the regular attention operator using

Table 5: Comparisons among MobileNetV2, SANet, and AttnNet in terms of top-1 accuracy, memory usage, and execution time. We report the actual memory usage using inputs of $128 \times 224^2 \times 3$. The time reported here is the model execution time of 100 iterations.

| Model | Top-1 | Memory | Time |
|---|---|---|---|
| MobileNetV2 | 0.720 | 7.4GB | 14.4s |
| SANet | 0.730 | 7.5GB | 14.9s |
| AttnNet | 0.730 | 11.7GB | 27.1s |

pooling operations. Note that AttnNet$_{1/N}$ and SANet achieve the same performance as AttnNet, which demonstrates the feasibility of replacing the expensive softmax$(\cdot)$ function with scaling by $1/N$. Notably, all networks with attention operators significantly outperform SENet which squeezes the spatial information. This demonstrates the effectiveness of attention operators that effectively build long-range relationships and lead to better performances.

### 4.4 ABLATION STUDIES

To fully demonstrate the effectiveness of our SAO, we conduct ablation studies based on MobileNetV2 and SANets. We add SAOs to MobileNetV2 to observe if it benefits other networks like MobileNetV2. We replace some inverted residual blocks with our AttnModules by referencing the structure of SANet, which we denote as MobileNetV2 w SAO. We remove SAOs from SANet by replacing AttnModule with BaseModule, resulting in SANet w/o SAO. We also explore the impact of the global item in SAO on the performance of SANets.

Table 6: Comparisons between MobileNetV2, MobileNetV2 with SAOs (MobileNetV2+SAO), SANet, SANet without SAO (SANet w/o SAO), and SANet without the global item in Eq. (5) in terms of the top-1 accuracy, and the number of total parameters on the ImageNet validation set.

| Model | Top-1 | Params |
|---|---|---|
| SANet | **0.730** | **3.44m** |
| SANet w/o SAO | 0.721 | 3.44m |
| SANet w/o global term | 0.728 | 3.44m |
| MobileNetV2+SAO | 0.727 | 3.47m |
| MobileNetV2 | 0.720 | 3.47m |

The comparison results are summarized in Table 6. By using SAOs, MobileNetV2 w SAO obtains a performance boost of 0.7% over that of the original MobileNetV2. The performance of

Table 7: Comparison results between U-Net, U-Net with regular attention operator (U-Net+Attn), and U-Net with SAO (U-Net+SAO) in terms of peak signal to noise ratio (PSNR), structural similarity index (SSIM), and normalized root mean square error (NRMSE). On the metrics followed by an ↑, the higher value indicates the better performance. Conversely, lower value indicates the better performance on the metrics followed by a ↓.

| Dataset | Metric | U-Net | U-Net+Attn | U-Net+SAO |
|---------|--------|-------|------------|-----------|
| Planaria | PSNR↑ | 31.571 | 31.951 | 31.929 |
| | SSIM↑ | 0.7707 | 0.7919 | 0.7898 |
| | NRMSE↓ | 0.0268 | 0.0257 | 0.0257 |
| Tribolium | PSNR↑ | 32.433 | 32.571 | 32.549 |
| | SSIM↑ | 0.9171 | 0.9201 | 0.9196 |
| | NRMSE↓ | 0.0241 | 0.0236 | 0.0239 |
| Flywing | PSNR↑ | 21.958 | 22.361 | 22.329 |
| | SSIM↑ | 0.5592 | 0.5903 | 0.5916 |
| | NRMSE↓ | 0.0798 | 0.0763 | 0.0767 |

SANet w/o SAO without using SAOs is 0.9% lower than that of SANet. Notably, SANet consumes the least computational costs while achieving the best performance. The results demonstrate that our SAO is consistently efficient and effective when being applied in different network architectures. Notably, it can be observed that the global item is important by comparing the performances between SANet and SANet w/o global term.

## 4.5 RESULTS ON 3-D IMAGE RESTORATION

In order to evaluate the effectiveness of our SAO in broader application scenarios, we conduct experiments on biological image restoration tasks, in particular, the 3D image denoising and the 3D image projection. The projection models map a noisy 3D image to a 2D plane, i.e., mapping 3D images from $\mathbb{R}^{h \times w \times c}$ to 2D images in $\mathbb{R}^{h \times w}$. We perform experiments on three different datasets collected by Weigert et al. (2018); those are Planaria, Tribolium and Flywing. Details of dataset are shown in Table 10 in the appendix. We adopt a general U-Net architecture (Ronneberger et al., 2015; Çiçek et al., 2016) as our baseline model. We add attention operators and our SAOs in bottom and decoder blocks. We apply linear transformations on $Q$, $K$ and $V$ in attention operators. The details of experimental setups are provided in the appendix. The comparison results are summarized in Table 7. Clearly, U-Net models with attention operators outperform the baseline model, which demonstrates the effectiveness of attention mechanism. Compared to U-Net with regular attention operators, SAOs result in similar results. This indicates our SAO is as effective as regular attention operators. Overall, these experimental results demonstrate that our SAO retains its effectiveness in broader application scenarios.

## 4.6 PARAMETER STUDY OF SAO

Since SAO involves extra trainable parameters in our Siamese similarity function, we study the impact of these trainable parameters in SANets. We use an all-ones vector to replace the trainable parameters $w$ in Eq. (6). We denote the resulting model as SANet w/o params. Table 8 reports the comparison results. We can observe that SANet outperforms SANet w/o params by a margin of 0.2% with regard to the top-1 accuracy using only 960 more parameters.

Table 8: Comparisons between SANet and the SANet removing trainable parameters in SAOs (SANet w/o params) in terms of the top-1 accuracy and number of parameters.

| Model | Top-1 | Params |
|-------|-------|--------|
| SANet | 0.730 | 3,449,408 |
| SANet w/o params | 0.728 | 3,448,448 |

This demonstrates that the trainable parameters in SAOs are necessary since the importance of different features are weighted.

## 5 CONCLUSIONS

In this work, we propose Siamese attention operators to overcome the excessive usage of computational resources of regular attention operators when being applied on high-order and/or high-dimensional data. We observe that the similarity score in the attention operator is computed using the dot product, which leads to high computational cost. To address this, we propose Siamese similarity function, which employs a feed-forward network to compute the similarity score between two input vectors. By using the shared network on two vectors, Siamese similarity function is symmetric. We use Siamese similarity function to replace dot product in regular attention operator, leading to Siamese attention operator. Theoretical analysis and experimental studies show that our SAOs significantly reduce the usage of computational resources. Based on SAOs, we design a family of efficient modules, leading to SANets. The evaluation on image classification tasks shows that our SANet significantly outperforms previous state-of-the-art models while using fewer trainable parameters and computational resources. In addition, we conduct experiments on 3-D image restoration to demonstrate the effectiveness of our SAOs in broader application scenarios. The parameter study shows that SAO brings great performance improvement with negligible extra trainable parameters.

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

# Appendix

## 1 THE SANETS ARCHITECTURE

Table 9 describes the SANets architecture. We use multiple heads with 4 heads in SAOs. The results of 4 heads are concatenated as the output. We only perform linear transformations on value matrices to save computational resources.

Table 9: Details of SANet architecture. Each line represents a sequence of operators and arguments in the format of "input size / operator name / expansion rate / output channels / number of operators / stride". We use "Conv2D", "AvgPool", and "FC" to denote 2D convolutional layer, global average pooling layer, and fully-connected layer, respectively. We use kernel size of $3 \times 3$ in depth-wise convolutional layers. The first depth-wise convolutional layer in a sequence of operators applies stride of $s$, while the other layers use stride of 1. We use $k$ to denote the class number in the task.

| Input | Operator | $r$ | $c$ | $n$ | $s$ |
|-------|----------|-----|-----|-----|-----|
| $224^2 \times 3$ | Conv2D $3 \times 3$ | - | 32 | 1 | 2 |
| $112^2 \times 32$ | BaseModule | 1 | 16 | 1 | 1 |
| $112^2 \times 16$ | BaseModule | 6 | 24 | 2 | 2 |
| $56^2 \times 24$ | BaseModule | 6 | 32 | 2 | 2 |
| $28^2 \times 32$ | AttnModule | 6 | 32 | 1 | 1 |
| $28^2 \times 32$ | BaseModule | 6 | 64 | 1 | 2 |
| $14^2 \times 64$ | AttnModule | 6 | 64 | 3 | 1 |
| $14^2 \times 64$ | AttnModule | 6 | 96 | 3 | 1 |
| $14^2 \times 96$ | BaseModule | 6 | 160 | 1 | 2 |
| $7^2 \times 160$ | AttnModule | 6 | 160 | 2 | 1 |
| $7^2 \times 160$ | AttnModule | 6 | 320 | 1 | 1 |
| $7^2 \times 320$ | Conv2D $1 \times 1$ | - | 1280 | 1 | 1 |
| $7^2 \times 1280$ | AvgPool + FC | - | $k$ | 1 | - |

## 2 EXPERIMENTAL SETUP FOR IMAGE CLASSIFICATION

We use the ImageNet ILSVRC 2012 image classification dataset in our experiments. We follow the data argumentation methods used in He et al. (2016b). During training, we scale the image into $256 \times 256$ and then randomly crop a $224 \times 224$ patch. Center cropping is used when performing inference. We use batch normalization to replace the scaling by $1/N$ in SAOs to enable a learnable scaling factor. All trainable parameters are initialized by Xavier initialization (Glorot & Bengio, 2010). Standard stochastic gradient descent optimizer with a momentum of 0.9 (Sutskever et al., 2013) is used to train the model for a total of 150 epochs. The learning rate is initially 0.1 and decays by 0.1 at the $80^{th}$, $105^{th}$, and $120^{th}$ epoch. We apply dropout (Srivastava et al., 2014) with a keep rate of 0.8 after the global average pooling layer. We use 8 GeForce RTX 2080 Ti GPUs with a batch size of 640 for training. Performance results on the validation dataset is reported since testing dataset labels are not available.

## 3 EXPERIMENTAL SETUP FOR IMAGE RESTORATION

The networks for the denoising models are the 3D U-Net with depth 3, which consists of two encoder blocks and two decoder blocks. The projection models consist of a decoder-encoder architecture that reduces 3D images into 2D tensors and a 2D U-Net. The number of training batches is 16 and the learning rates are 0.0004 for all the three tasks. The denoising models are trained for 200 epochs while the projection model is trained for 100 epochs. The summary of the datasets used in this task is provided in Table 10.

Table 10: Summary of datasets used in our image restoration experiments. Planaria and Tribolium are used for image denoising experiments and Flywing is used in image projection experiments.

| Dataset | Task | # Training | Patch Size | # Testing | Testing Size |
|---|---|---|---|---|---|
| Planaria | Denoising | 17,005 | $16 \times 64 \times 64 \times 1$ | 20 | $95 \times 1024 \times 1024$ |
| Tribolium | Denoising | 14,725 | $16 \times 64 \times 64 \times 1$ | 6 | $50 \times 800 \times 800$ |
| Flywing | Projection | 16,891 | $50 \times 64 \times 64 \times 1$ | 26 | $50 \times 520 \times 692$ |

## REFERENCES

All citations refer to the references in the main paper.

