# OpenReview forum: "Siamese Attention Networks"
_ICLR.cc/2020/Conference — Reject_

### Official Review · AnonReviewer3 · 2019-10-23
**Official Blind Review #3**

**Rating:** 6

**Review:**

SUMMARY: A new similarity function replacing the dot product of key and query in attention modules - instead take a shared-weighted sum

Good paper, sound theory, very clear explanations. Literature review was sufficient to explain the problem and underlying theory.

Results look convincing, although they cannot be verified unless code is shared.

Reasonable direction of exploration - there are several possible similarity functions, this paper explores one of them that offers significantly less computational resources, which are essential for on-device applications. Thorough exploration of this idea was done and I am convinced this is a good alternative to regular attention.

All the rest of the paper was to incorporate this new method in different tasks in different architectures using or not using attention, and seeing the differences in terms of computational resources. Well thought experiments and results. Again, cannot be verified unless code is shared.

**Experience Assessment:**

I have read many papers in this area.

**Review Assessment: Checking Correctness Of Derivations And Theory:**

I assessed the sensibility of the derivations and theory.

**Review Assessment: Checking Correctness Of Experiments:**

I assessed the sensibility of the experiments.

**Review Assessment: Thoroughness In Paper Reading:**

I read the paper at least twice and used my best judgement in assessing the paper.

---

> ### Author Response · Authors · 2019-11-06
> **Code release**
>
> Thank you for your comments. We will release our code in the final version.

---

### Official Review · AnonReviewer1 · 2019-10-23
**Official Blind Review #1**

**Rating:** 3

**Review:**

The authors introduce a novel self-attention operator for neural networks. Their self-attention operator computes similarity between elements a and b as (a+b)^Tw where w is a learned parameter and does not use the softmax operator. This leads to improvements in space and time complexity compared to regular self-attention which uses the dot product (a^Tb).
They show that concatenating their operation with convolution brings improvements over the MobileNetv2 baseline on ImageNet classification and over U-Net on restoration tasks.

Attention has been empirically shown to bring improvements in many visual tasks but certain methods (such as self-attention) can be quite expensive in computations and memory. Identifying cheaper attention mechanisms that obtain similar accuracy performance as expensive attention mechanisms is therefore an important direction for work.

However, I take several crucial issues with this work and especially the evaluation/presentation of the methods:
- Although this is the focus of the work, the authors do not report actual memory consumption and latency times for MobileNetv2, SANet and the other attention mechanisms. (There is no need for the simulated scenarios of Table 2 since we already know the theoretical complexities of the different methods).
- The authors only compare their methods against regular self-attention (without or with pooling/softmax), and ignore a longstanding literature of other (potentially cheaper in terms of memory and computations) attention mechanisms in vision (see below). Without comparison to at least Squeeze-and-Excite, it is hard to evaluate the significance of the method presented in the draft.
- The motivation for naming the method "siamese" is quite poor. Siamese networks typically are more complex than a single layer feed forward (which is just a dot-product). Furthermore, the siamese similarity (as introduced by the authors) does not respect the usual properties of similarity functions. For example siasim(a, 0) = a^tw = 1/2 siasim(a,a) can take arbitrary values including negative values ("a can be dissimilar with itself")
- X vs Q, K, V? Self-attention is incorrectly described as "a special case of the attention operator with Q = K = V", instead of Q = XW^Q, K=XW^K, V = XW^V.
- In Table 7, shouldn't SANet w/o params have less params than SANet?

In summary, the paper addresses an important challenge and proposes a technically sound method.  However, the current  draft has fundamental experimental flaws in its evaluation/presentation and lacks comparison against relevant cheap channelwise attention mechanisms (such as Squeeze-and-Excitation). I argue for rejection.

Relevant literature:
- channelwise attention: Squeeze-and-Excitation, Gather-Excite
- Channelwise and spatial attention: Bottleneck Attention Module, Convolutional Block Attention Module
- Relative Self-attention for vision: Attention Augmented Convolutional Networks, An Empirical Study of Spatial Attention Mechanisms in Deep Networks.


**Experience Assessment:**

I have published one or two papers in this area.

**Review Assessment: Checking Correctness Of Derivations And Theory:**

I carefully checked the derivations and theory.

**Review Assessment: Checking Correctness Of Experiments:**

I carefully checked the experiments.

**Review Assessment: Thoroughness In Paper Reading:**

I read the paper thoroughly.

---

### Official Review · AnonReviewer2 · 2019-10-25
**Official Blind Review #2**

**Rating:** 6

**Review:**

In this paper, the authors propose a new mechanism to perform the attention operators. The similarity between a key and a query is performed as the dot product between a trainable weight and the addition of the key and query.  The proposed Siamese attention operator is much more efficient than prior attention methods in terms of speed. The evaluation on a few computer vision tasks shows the presented method performs as well as the typical attention methods, but it runs much faster.

First of all, I think this attention method should be quite useful for various neural networks. It is faster and performs equally well as other attention operators.  However, I do have some concerns about this method.

- It is not clear to me why this method works. (a+b)^T*w is a strange expression to compute the similarity between a and b. No much explanation or intuition is given in the paper, and I have no clue why this works.

- It seems to me that the proposed method performs slightly worse than the regular attention, according to Figure 6.

My overall rating is borderline. It would be great if the authors can resolve my concerns.

Other questions:
- Why SANet (MobileNetv2+SAO) has fewer parameters than MobileNetv2 in Table 3?
- Is attention an operator that significantly slows the speed of the whole network? If not, the speedup of attention is not that important.


**Experience Assessment:**

I have read many papers in this area.

**Review Assessment: Checking Correctness Of Derivations And Theory:**

I assessed the sensibility of the derivations and theory.

**Review Assessment: Checking Correctness Of Experiments:**

I assessed the sensibility of the experiments.

**Review Assessment: Thoroughness In Paper Reading:**

I read the paper at least twice and used my best judgement in assessing the paper.

---

### Author Response · Authors · 2019-11-12
**Comment on the new submission**

Dear reviewers,

We have updated our submission with following changes:

1. We conduct experiments to compare our methods with Squeeze-and-Excite.

2. We report the actual memory usage and time consumption of MobileNetV2, SANet, and AttnNet.

3. We correct an error in Table 7.

---

### Decision · Program_Chairs · 2019-12-19

**Decision:**

Reject

**Comment:**

The submission presents a Siamese attention operator that lowers the computational costs of attention operators for applications such as image recognition. The reviews are split. R1 posted significant concerns with the content of the submission. The concerns remain after the authors' responses and revision. One of the concerns is the apparent dual submission with "Kronecker Attention Networks". The AC agrees with these concerns and recommends rejecting the submission.